# The Impact of COVID-19 and Climate Change on Food Security in Pamijahan District, Bogor Regency

**Frema Apdita \*, Johan Iskandar and Emma Rochima \***

Regional Innovation Graduate School, Universitas Padjadjaran, Bandung 45363, Indonesia; johan.iskandar@unpad.ac.id
\* Correspondence: frema21001@mail.unpad.ac.id (F.A.); emma.rochima@unpad.ac.id (E.R.)

**Abstract:** Food security is a requirement for meeting household food demands and is expressed in the availability of enough food that is sufficient both in quantity and quality, safe, equitable, and inexpensive. Academics and practitioners have attempted to revise food security models that may depict disaster-prone places, particularly Pamijahan District; however, these varied models each have their setbacks when compared to the world's various global conditions. This study aims to examine how food security is affected by the availability, accessibility, and consumption of food under the influence of climate change and the COVID-19 outbreak in the period 2017–2022. The methods used in this study were mixed-methods (quantitative and qualitative). In this study, participants underwent SMART PLS 3.0 analysis, followed by quantitative analytic techniques. Study results showed that the total food security condition of Cibunian Village in Pamijahan District in the period 2017–2022 can be categorized as vulnerable. Based on the FSVA analysis, it revealed that Cibunian Village was in the category of being vulnerable to food insecurity in general for the 2017–2022 period, while based on the SKPG analysis from the perspective of food access, there has been a 33.3% increase in food insecurity. The COVID-19 outbreak, climate change, and food consumption are the causes, and they all significantly and positively affect food security. This work advances our knowledge of food security in the COVID-19 outbreak age and the issues posed by global climate change. Everywhere, even in disaster-prone areas, complete food security should be attained.

**Keywords:** climate change; COVID-19; food security; food availability; food accessibility; food utilization

## 1. Introduction

Food is a human right and becomes a government obligation to fulfill (Dohlman et al. 2019). Currently, the world is facing climate change due to global warming. Climate change has increased the frequency of hydrometeorological disasters (Azizah et al. 2022). Since the last decade, climate change has become a major threat to food security.

Indonesia is located on the equator and is influenced by various weather and climate extremes. Variations in weather and climate affect forecasts of planting and harvesting seasons, seed supply distribution systems, and crop yields, causing potential problems in the food security system (Sakya and Mahardh 2010).

The Corona Virus Disease (COVID)-19 outbreak at the end of 2019 exacerbated the challenges of achieving food security. The COVID-19 outbreak created a multidimensional crisis in the health, economic, social, and political sectors, with further implications for food security (Rahmah et al. 2020). The impacts of climate change and the COVID-19 outbreak require further analysis to achieve sustainable food security (Rasul 2021). Achieving sustainable food security to end hunger refers to the SDGs (Sustainable Development Goals) program's 2nd (zero hunger) and 13th (climate action) goals.

Food security is closely related to food availability and food access (Mun'im 2021). Two global crises, the COVID-19 outbreak and climate change, demand appropriate action

from the government so that food security can be realized (Molden et al. 2020). On the other hand, hydrometeorological situations and the COVID-19 outbreak are creating instability in food security.

Food security is mandatory around the world, including in Indonesia. The Economist Group (2022) reported that Indonesia's Food Security Index was ranked 63rd out of 113 countries in 2022. According to the report, the sustainability and adaptation pillars have the lowest indicator performance in the index. They show that productive research related to the development of food security that is adaptive to climate change risks is still seriously needed in Indonesia, including in Bogor Regency.

Alinovi et al. (2010) state that resilience is influenced by stability, social safety nets, access to public services, assets, income, access to food, and adaptive capacity. Alam et al. (2016) stated that the food security model consists of food availability, food accessibility, food utilization, and food security.

Bogor Regency is a second-level administrative region of West Java Province, Indonesia, and is very important as one of the buffer zones for the capital city of Jakarta. According to the Statistical Agency of West Java Province, 6,088,233 people lived in Bogor Regency in 2020, representing 12.19% of the entire population of West Java Province (Yudhanto et al. 2023).

Bogor Regency's Food Security Index is ranked 317 out of 417 districts in Indonesia. Bogor Regency is included in a disaster-prone area. In 2021, the Bogor Regency Regional Disaster Management Agency reported that the frequency of disasters in Bogor Regency from 2017–2020 had continued to increase.

The Food Security Vulnerability Atlas (FSVA) is a thematic map that visualizes the geographical conditions of food insecurity (Food Security Agency 2022). In 2021, the FSVA of Bogor Regency, Pamijahan District, categorized food security, and especially food availability, as vulnerable in terms of access and utilization of food.

The uniqueness of Pamijahan District, surrounded by the Salak Mountain area, is that it has high potential for food resources and also high disaster vulnerability (Regional Disaster Management Agency 2022). This area is located in a dangerous zone. In the period 2020–2022, La Nina winds will come to this area and threaten potential disasters such as floods and landslides. It was recorded that on 22 June and 23 June 2022, flash floods and landslides occurred in Pamijahan District, Bogor Regency, which resulted in 194 heads of families being affected in Cibunian Village and Purwabakti Village (Regional Disaster Management Agency 2022)

Pamijahan District is located at coordinates 106°38′00″ to 106°42′00″ East longitude and 6°38′00″ to 6°44′00″ South latitude. The slope of the surrounding area ranges from 8% to 40%, and the height of the land in Pamijahan District is in the range of 1000–2000 m (Ulfah Rahayu et al. 2019).

The average temperature in Pamijahan District in the period 1991–2002 was 25.60 °C, with an increase in temperature from 1991–2022 of 0.60 °C and the highest rainfall reaching more than 500 mm per month. It can be categorized as an area with an extreme climate (BPS 2021).

From 1991 until 2022, there were 46 hydrometeorological disasters, which were dominated by floods, landslides, and strong winds. The high incidence in a region will be affected by climate change, increasing the potential for food insecurity in that area. The locations for collecting data for this study were Cibunian Village and Purwabakti Village in Pamijahan District because they are categorized as very disaster-prone villages with high rainfall and steep slopes (Ulfah Rahayu et al. 2019).

Facing various existing obstacles, residents of Pamijahan District must have the ability to maintain their food security for survival in the future. Based on this experience, a model of food security in disaster-prone areas, especially those affected by climate change and those that have also experienced non-natural disasters such as COVID-19, is an interesting thing to develop for the sustainability of the area. This food security model is a planning

and prediction technique that has the advantage of dealing with opportunities for similar events in the future.

Food security as a way to deal with hunger is strongly supported by food availability, food access, and food utilization. The support of these three sub-systems must be strong so that they can face various interventions from outside and within the system. Several previous studies have not formulated a model of food security that has been analyzed statistically; instead, they are still conceptual in nature, not referring to empirical studies.

Through this phenomenon, researchers have an interest in conducting empirical studies by identifying the unique characteristics of residents who live in disaster-prone areas and measuring the relationship between model variables of food security that are related to climate change disasters and the COVID-19 outbreak.

Good food security planning can secure the continuity of food availability, access to food, and utilization of food for residents of Pamijahan District. One of the approaches used to deal with external interventions is to develop a model. This research investigates the characteristics of the residents of Pamijahan District and the food security model that was formed to maximize the potential for food availability, food access, and food utilization in the face of climate change and outbreaks of non-natural disasters.

The results of this study will be useful for developing strategies for the government to always maintain suitable food security in disaster-prone areas. In this study, structural equity modeling through partial least squares (PLS-SEM) is also used in the new field of food security.

The appropriate statistical technique for predicting food security management planning in disaster-prone areas is partial least squares structural equation modeling (PLS-SEM). This technique is used because it prioritizes predictive results without requiring normal distribution assumptions, and this technique is very good to use when the sample size is small (Joseph F. Hair et al. 2019).

The PLS-SEM analysis tool is Smart PLS. The use of Smart PLS is highly recommended when you have a limited number of respondents and the model being built is complex. In this study, Smart PLS Series 3.0 is used because this research is predictive and explains latent variables rather than testing a theory with a small number of samples (J. H. Hair et al. 2017).

## 2. Materials and Methods

A questionnaire with a sampling method was used to determine the condition of residents in Pamijahan District. Data collection, surveys, direct distribution of questionnaires to respondents, and in-depth interviews with an expert are all parts of the research method.

The research location is Pamijahan District, which supplies food in Bogor Regency (Indonesian Ministry of Agriculture's Food Security Agency 2021). The uniqueness of this district is its susceptibility to food security due to flash floods and landslides in June 2022. Figure 1 provides a map of the area studied in Pamijahan District, Bogor Regency.

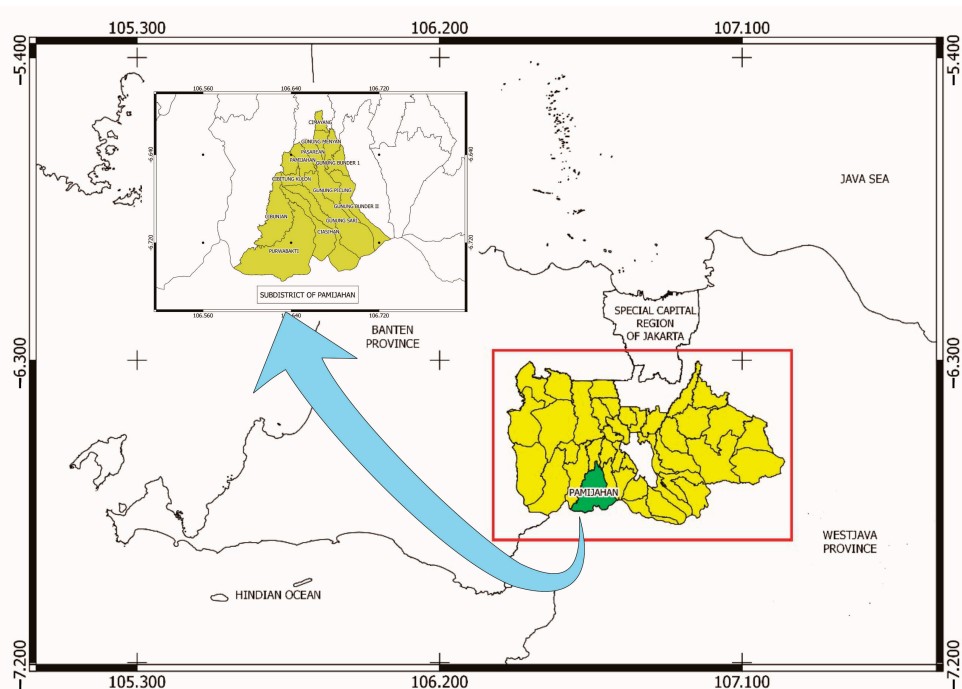

**Figure 1.** Pamijahan District map of Bogor Regency.

## 2.1. Data Collection Method

In analyzing food security above, researchers combined primary and secondary data. Primary data was collected through surveys using questionnaires and in-depth interviews to determine residents' understanding of the effects of climate change and the COVID-19 outbreak on food security. This study was carried out in Pamijahan District, Bogor Regency, with respondents who were community members who were very affected by the climate change disaster and the COVID-19 outbreak. Secondary data was obtained from the various relevant literature, journals, books, and government statistical data.

The purpose of collecting data through a questionnaire is to access the sub-system of food security (food availability, food access, and food utilization) regarding the presence of climate change and non-natural disasters (COVID-19). The variable measurement of food availability was developed by 2021, while food access was introduced by Béné et al. (2021). Food utilization is measured using the approach from (Baliwati 2019), the climate change variable is measured based on the explanation from (Rasul 2021), and the COVID-19 outbreak is measured using variables from (Hendriks et al. 2022).

The process of collecting information for this study was carried out by interviewing relevant stakeholders. As for the observation process, the authors carried it out directly with respondents. Detailed research stages can be seen in Figure 2.

In Figure 2, this study begins with observing the food security system, which consists of food availability, access, and utilization (Alam et al. 2016). The second step is to determine the food security model with the variables of climate change and the COVID-19 outbreak so that a design model for food security can be produced in disaster-prone locations. The third stage is interviews with respondents and in-depth interviews with key persons using questionnaire media (this research questionnaire can be seen in Appendix A), as well as an interview guide, so that a reflection of the condition of food security is produced, which is affected by climate change and the COVID-19 outbreak. The results of the path analysis using SEM-PLS and the derived food security situation from the results of the FSVA and SKPG analyses, compiled with the results of in-depth interviews with key persons, formulated managerial implications for achieving food security in disaster-prone areas.

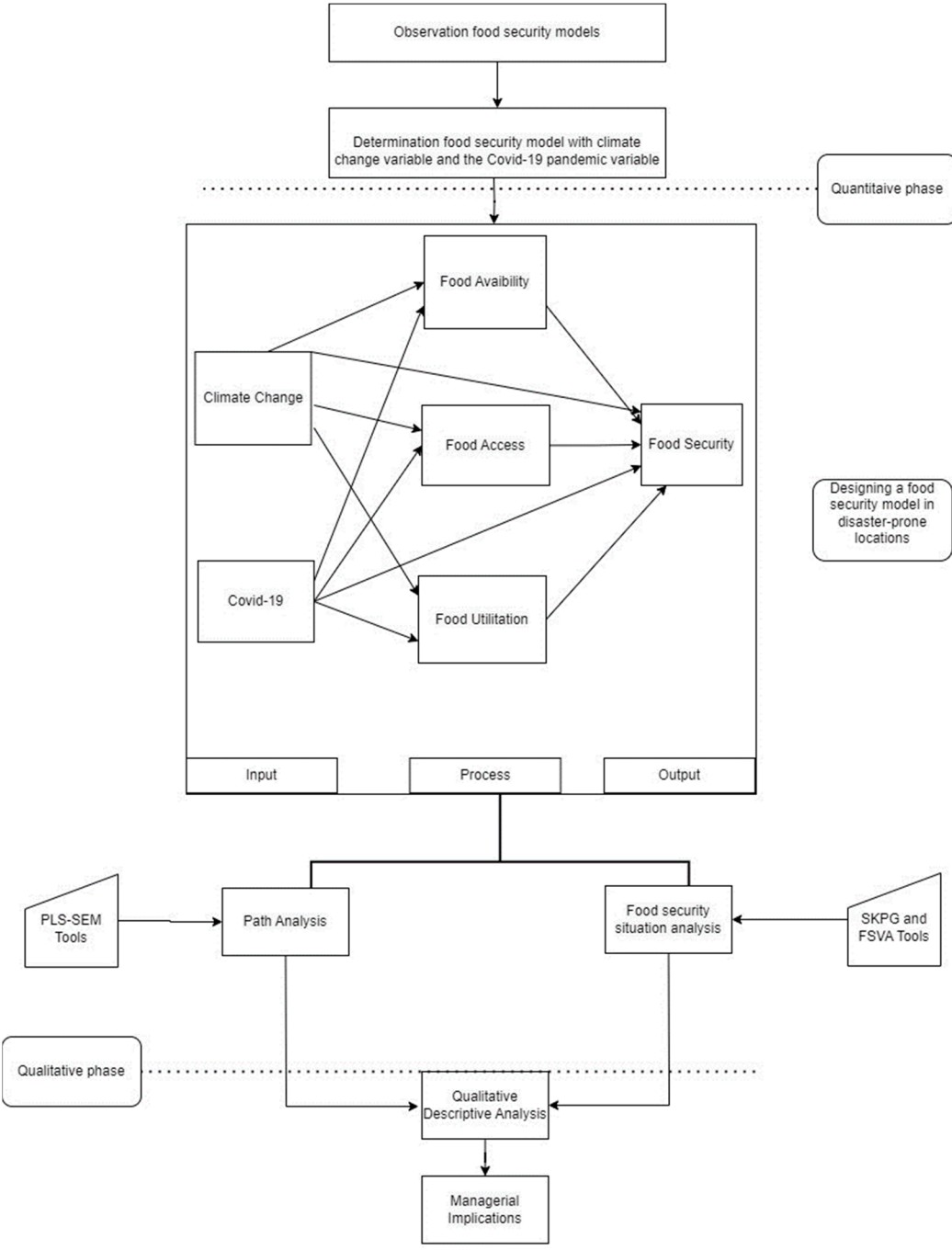

**Figure 2.** Research method.

All items were evaluated using a Likert scale of one to five, with five expressing strong agreement. The Likert scale is used to convey how strongly respondents agree or disagree with specific statements about actions, things, people, or events. The suggested scale typically consists of five points. A Likert scale was chosen with five class scores as the measurement. There are a total of five groups, made up of the average value of each informant. The following formula can be utilized to determine class intervals:

Explanation:

$SR$ = Range (0.8)

$$SR = \frac{(a-b)}{c}$$

$b$ = Minimum scores (1)

*a* = Maximum scores (5)
*c* = Number of class intervals (5)

These calculations enable us to establish that the calculated scale range is 0.8. According to the statement on the research questionnaire, the average range of 1.00–1.80 can be categorized into the Poor category; >0.80–2.60 can be categorized into the Fair category; >2.60–3.40 can be categorized into the Good category; >3.40–4.20 can be categorized into the Very Good category; and >4.20–5.00 can be categorized into the Excellent category. The items used to measure each variable are listed in Table A1 in Appendix A.

### 2.2. Data Analysis

The data processing and analysis for this study employed partial least squares structural equation modeling, validity testing, reliability tests, descriptive analyses, and simulation of partial least squares structural equilibrium (PLS-SEM). In this study, Smart PLS 3.0 was used. Descriptive analysis was performed to obtain a wide description of the characteristics of respondents, including gender, age, education, and respondent profile, as well as to describe food security implementation using the mean. For the measurement, a 5-point Likert scale was used to determine the scale range. A Likert scale is used to assess respondents' attitudes, views, and perceptions of social issues.

Secondary data in the form of FSVA maps and SKPG maps of Pamijahan District from the Bogor District Government were analyzed in relation to the food security situation in the Bogor District. Primary data for quantitative analysis was collected using a survey method with questionnaire aids, as well as the results of in-depth interviews with key persons in Pamijahan District who understand the characteristics and local ecological knowledge that can maintain food security in the area, as well as residents and housewives who have babies or toddlers who are affected by floods and the COVID-19 outbreak. The population of this study was composed of 194 families affected by natural disasters in Pamijahan District, Bogor Regency. The locations affected by the disaster in Pamijahan District are Cibunian Village and Purwabakti Village. The determination of examples for research respondents is based on the formula from Frank Lynch as follows (Iskandar 2018):

$$n = \frac{NZ^2.p(1-p)}{ND^2 + Z^2.p(1-p)}$$

where:

*n* = Sample size (64)
*N* = Total population (194)
*Z* = Value in the area under the normal distribution curve (1.96)
*P* = Largest possible proportion (0.50)
*D* = Degree of deviation (10%)

By using a population-sample table based on the formula from Frank Lynch, for a total population (N) of 194 at a 95% confidence level and a degree of deviation (D) of 10%, the sample size (n) in this study amounted to 64.

### 2.3. Model Analysis

The study began with an analysis of the food security situation in Pamijahan District, Bogor Regency, using secondary data from the results of the FSVA analysis and SKPG analysis from the Bogor Regency Food Security Service (Rimadianti et al. 2016). Termination of the indicators that most influence the status of food security after the disaster due to climate change and COVID-19 in Pamijahan District, Bogor Regency, uses path analysis with Smart PLS 3.0 software (Yudhanto et al. 2023). Then the results of the analysis above are compiled with the results of in-depth interviews to formulate managerial implications for maintaining food security in disaster-prone areas. This research uses a validity test, a reliability test, descriptive analysis, and partial least squares structural equation simulation (PLS-SEM).

In FSVA analysis, there was a change in the assessment indicators, which were originally nine indicators in 2017, to six indicators in 2019. The nine FSVA indicators in 2017 include the ratio of stalls to households, the ratio of shops to households, the ratio of people with the lowest welfare status, the ratio of households without access to electricity, the number of villages without adequate connecting access, the ratio of children not attending school, the ratio of households without access to clean water, the ratio of health workers to residents, and the ratio of households without residential facilities. Then the FSVA analysis was refined into six indicators covering the ratio of agricultural land area, the ratio of food supply infrastructure, the ratio of people with the lowest level of welfare, the number of villages with inadequate connecting access, the ratio of households without access to clean water, and the ratio of the number of villagers per health worker. The six indicators are weighted according to their level of importance, then a composite score is calculated and grouped according to the cut-off for each category.

The measurement model (outer model) and the structural model (inner model) are the two sub-models that make up the PLS-SEM analysis (Joe F. Hair et al. 2014). Construct convergent validity, discriminant validity, and reliability were assessed using the outer model (J. H. Hair et al. 2017). In addition, the inner model is used to assess the relevance of the path coefficient and R-square value. Two categories of variables are used in PLS-SEM. The first is the observed variable, sometimes known as the manifest variable because it can be viewed immediately. The second category is unobserved variables, sometimes known as latent variables because they cannot be observed directly (Joe F. Hair et al. 2014). Together with the six latent variables, there are 112 manifest variables. The model of this study can be seen in Figure 3. Previously, a content validation test was carried out on the research questionnaire to obtain validity (Ayu Dessy Sugiharni 2018), conducted by two experts, including academics and practitioners in the field of food security. The value of the content validity ratio (CVR) from the results of the questionnaire content validation test in this study was 0.89, so the questionnaire can be said to be valid.

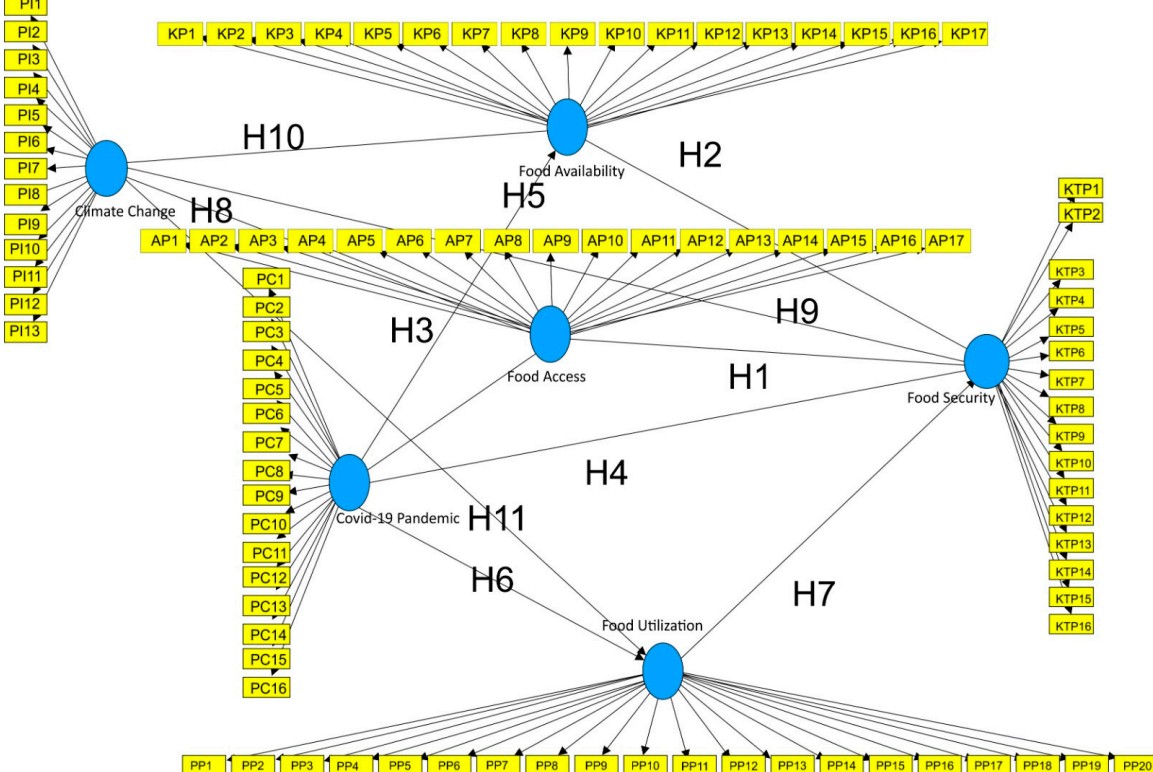

**Figure 3.** Research models.

The hypotheses in this study can be described as follows:

**H1.** *Access to food has a positive and significant effect on food security in Pamijahan District.*

**H2.** *Food availability has a positive and significant effect on food security in Pamijahan District.*

**H3.** *The COVID-19 outbreak has had a positive and significant effect on access to food in Pamijahan District.*

**H4.** *The COVID-19 outbreak has had a positive and significant effect on food security in Pamijahan District.*

**H5.** *The COVID-19 outbreak has had a positive and significant effect on food availability in Pamijahan District.*

**H6.** *The COVID-19 outbreak has had a positive and significant effect on food utilization in Pamijahan District.*

**H7.** *Food utilization has a positive and significant effect on food security in Pamijahan District.*

**H8.** *Climate change has a positive and significant effect on access to food in Pamijahan District.*

**H9.** *Climate change has a positive and significant effect on food security in Pamijahan District.*

**H10.** *Climate change has a positive and significant effect on food availability in Pamijahan District.*

**H11.** *Climate change has a positive and significant effect on food utilization in Pamijahan District.*

Outer model evaluation and inner model evaluation are two evaluation models used in PLS-SEM data analysis (Cheung 2013). The external model is used to test the effect of latent variable indicators. Multicollinearity was used in this work to clarify the data without any visible bias before analysis. The absence of multicollinearity problems is a prerequisite for properly checking the outer model. Situations with substantial correlations or connectedness between indicators are called multicollinearity. A variance inflating factor (VIF) value of more than five indicates a multicollinearity correlation value, which is defined as a correlation value of more than nine. Multicollinearity occurs if the VIF value of the latent variable is greater than five. Actions that can be taken include reducing or eliminating indications with a high degree of association (J. H. Hair et al. 2017).

The evaluation of the outer model consists of three tests. The convergent validity test can be used to assess how well the manifest variable can explain hidden variables by looking at the loading factor above 0.50. When the average variance extract (AVE) results are greater than 0.50, the discriminant validity test is used to assess how much the latent and manifest variables differ from each other. Previous research explained the relationship between Cronbach's alpha above 0.60 and the composite reliability used to test composite reliability (J. H. Hair et al. 2017). The inner model is used to determine the effect of the independent variables on the dependent variable by comparing the coefficient of determination (R square) and the path coefficient (Ghozali and Latan 2015).

## 3. Results and Discussion

### 3.1. Food Security Conditions in Pamijahan District

The conceptual framework for regional food security considers food availability, access to food, and utilization of food, which guarantees that all individuals have the right to obtain food according to their needs. The opposite condition of food security is called food and nutrition insecurity.

Information related to food insecurity was analyzed using two analytical tools according to its causes. Based on the causes, food insecurity can be divided into chronic food insecurity and transient food insecurity. Chronic food insecurity was analyzed using the analysis map of Food Security and Vulnerability, or Food Security and Vulnerability Atlas (FSVA), and transient food insecurity using the Food and Nutrition Alertness System/SKPG analysis.

### 3.1.1. FSVA Analysis

The FSVA analysis for Pamijahan District was carried out in 2017, 2019, 2021, and 2022. Table 1 shows the composite results of the FSVA analysis for 2017, 2019, 2021, and 2022 in the Pamijahan District.

**Table 1.** FSVA analysis of Cibunian Village and Purwabakti Village.

| Village | Year | | | |
|---|---|---|---|---|
| | **2017** | **2019** | **2021** | **2022** |
| Cibunian | 1 | 4 | 3 | 3 |
| Purwabakti | 1 | 3 | 2 | 3 |

Source: Compiled by the author

The results of the FSVA analysis were divided into 2 groups, namely the food-insecure vulnerable group consisting of Priority 1 (very food-insecure), Priority 2 (food-insecure vulnerable), and Priority 3 (somewhat vulnerable to food insecurity). The food-secure group consists of Priority 4 (somewhat food-secure), Priority 5 (food-secure), and Priority 6 (very food-secure).

In 2017, Cibunian Village and Purwabakti Village were in Priority 1, namely in the vulnerable category of food insecurity caused by the low level of welfare of the population, the high number of children who are not in school, and the high number of households that do not have clean water facilities. There are differences in the approach and method of calculating the 2017 FSVA analysis, so it cannot be compared with the following year's FSVA analysis.

### 3.1.2. SKPG Analysis

SKPG is an early warning system adopted from GIEWS, the Global Information and Early Warning System on Food and Agriculture (Shaw 2007). In the SKPG analysis, three aspects are used: the availability aspect (planting area and puso area), the aspect of food access (food prices), and the aspect of food utilization (nutritional status of toddlers). This study found an increase in food insecurity in terms of food availability and food access. Figure 4 presents the development of the results of the availability aspect analysis in Pamijahan District in the 2017–2022 period.

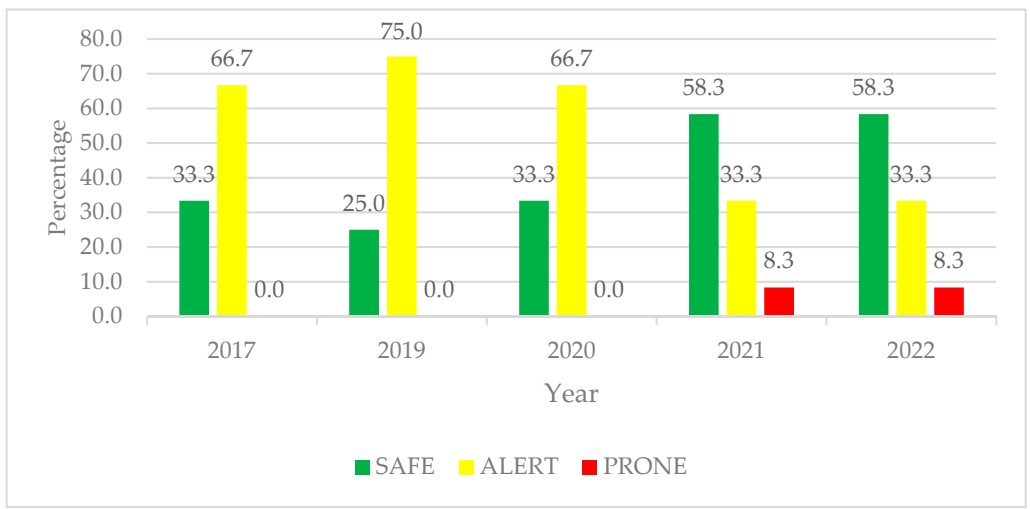

**Figure 4.** Food availability in 2017–2022.

Based on the SKPG analysis, the food security status is classified into three categories: safe, alert, and food insecure. Food security generally looks stable in terms of availability in 2017, 2019, 2021, and 2022, but there has been an 8.3% increase in food insecurity. This

shows an increase in crop failures caused by natural disasters such as floods, droughts, and plant-destroying organism attacks as a result of climate change (UNFCC 2015). Figure 5 shows the situation of food security in terms of food access.

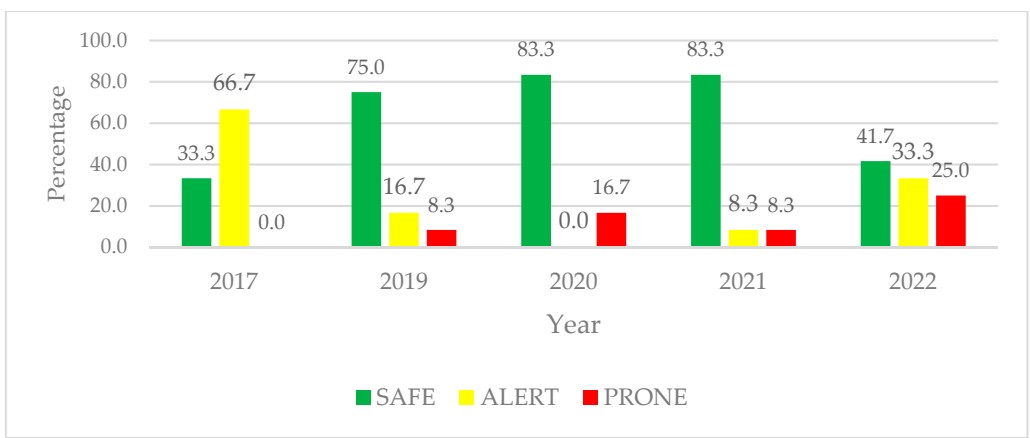

**Figure 5.** Access to food in 2017–2022.

Changes in rice prices compared to prices in the previous three months are used to assess food access. The vulnerable category increased from 0% to 25% from 2017 to 2022, while food alert conditions increased from 2017 to 33.3% in 2022, indicating an increase in food costs beginning in 2019 owing to the COVID-19 pandemic. The COVID-19 epidemic poses a significant danger to food availability (Rume and Islam 2020). The COVID-19 pandemic has caused a 20% increase in global food prices (Laborde et al. 2020).

*3.2. Determination of the Most Influential Indicators on Food Security in Pamijahan District, Bogor Regency*

3.2.1. Common Method Bias

A problem known as common method bias (CMB) occurs when the measurement technique used in SEM studies causes problems rather than a network of causes and effects among the latent variables in the model being investigated (Kock 2015). In this study, Smart PLS is used to identify CMB threats. The test indicates that the VIF element is lower than the 3.3 threshold. This shows that the model is free from CMB (J. H. Hair et al. 2017; Kock 2015).

3.2.2. Model Measurement

The suitability of the measurements is checked using validity and reliability standards. The ability of a measuring device (or object) to consistently produce the same result is known as reliability. Validity is a measure of how accurately an understanding is measured by measuring instruments (items). Because there is a multicollinearity condition, actions that can be taken include reducing or eliminating indications with a high degree of association. The VIF/correlation matrix measurement results at the manifest variable level for all latent variables in Table 1 are listed below, while the summary of model measurements after the multicollinearity test is shown in Table 2.

The assumption that must be met when analyzing the outer model is that there are no multicollinearity problems. Multicollinearity is a problem of interconnection or strong correlation between indicators. The multicollinearity correlation value is indicated by a correlation value of more than 9, which is indicated by a variance inflating factor (VIF) value of more than 5.

**Table 2.** Correlation matrix information.

| Items | AP | KTP | KP | PC | PP | PI |
|---|---|---|---|---|---|---|
| AP | | 2.023 | | | | |
| KTP | | | | | | |
| KP | | 1.096 | | | | |
| PC | 1.194 | 1.510 | 1.194 | | 1.194 | |
| PP | | 1.739 | | | | |
| PI | 1.194 | 1.509 | 1.194 | | 1.194 | |

Source: Compiled by the author. Remarks: AP: access to food; PC: COVID-19 outbreak; ETC: food security; PP: utilization of food; KP: food availability; PI: climate change.

If there is a latent variable VIF value of more than 5, then there is multicollinearity. The consequent actions that can be taken include dropping or removing indicators that have a strong correlation. Following are the results of VIF measurements at the manifest variable level for all latent variables shown in Table 3.

**Table 3.** Summary of model measurements after the multicollinearity test.

| Item | Indicator | Measurement Result | | | | | | Supported |
|---|---|---|---|---|---|---|---|---|
| Outer Loading | >0.7 | PI4 | 0.874 | KP4 | 0.821 | KP3 | 0.783 | Yes |
| | | PC15 | 0.816 | AP4 | 0.760 | AP1 | 0.754 | |
| | | KP8 | 0.796 | KTP1 | 0.799 | PP5 | 0.811 | |
| | | PP2 | 0.896 | PC13 | 0.872 | PC7 | 0.833 | |
| | | PI5 | 0.884 | KP5 | 0.796 | PP1 | 0.853 | |
| | | PC16 | 0.810 | AP9 | 0.874 | KTP12 | 0.900 | |
| | | KP9 | 0.816 | KTP11 | 0.886 | PC6 | 0.732 | |
| | | PP3 | 0.829 | PC14 | 0.830 | KP6 | 0.838 | |
| Average Variance Extracted (AVE) | >0.5 | PI | | 0.773 | | PC | 0.667 | Yes |
| | | KP | | 0.649 | | AP | 0.636 | |
| | | PP | | 0.718 | | KTP | 0.734 | |
| Composite Reliability | >0.6 | PI | | 0.872 | | PC | 0.923 | Yes |
| | | KP | | 0.937 | | AP | 0.839 | |
| | | PP | | 0.911 | | KTP | 0.892 | |
| Cronbach Alpha | >0.6 | PI | | 0.901 | | PC | 0.901 | Yes |
| | | KP | | 0.924 | | AP | 0.731 | |
| | | PP | | 0.871 | | KTP | 0.816 | |

Source: Compiled by the author.

All of the statements in the questionnaire were declared valid at the 5% significance level, where the r count exceeded the r table based on the results of the validity and reliability of 64 samples (0.361). In this study, the value of Cronbach's alpha for each variable was greater than 0.06, which indicates the dependability of the variable.

The Fornell-Larcker criterion, measuring the degree of anticipated "difference" between items for various factors, was used to test for discriminant validity. The square of the correlation was compared with the AVE of each factor to assess the discriminant validity of the model. In the other case, the correlation coefficient between factors is considered to have very good discriminant validity when the AVE is greater than the correlation coefficient between factors and other factors. The values on the diagonal represent the square root of AVE (J. H. Hair et al. 2017), while the values outside the diagonal are correlations. The results of discriminant validity are shown in Table 4.

**Table 4.** Discriminant validity matrix.

| Items | AP | KTP | KP | PC | PP | PI |
|---|---|---|---|---|---|---|
| AP | 0.798 | | | | | |
| KTP | 0.429 | 0.857 | | | | |
| KP | 0.259 | 0.263 | 0.805 | | | |
| PC | 0.551 | 0.317 | 0.149 | 0.816 | | |
| PP | 0.597 | 0.457 | 0.254 | 0.455 | 0.848 | |
| PI | 0.526 | 0.522 | 0.104 | 0.403 | 0.489 | 0.879 |

Source: Compiled by the author.

### 3.2.3. Result Analysis

With a loading factor value of 0.50 and no multicollinearity problems, 26 of the 112 tested indicators passed the convergent validity test, according to the outer model assessment (Table 1). In the discriminant validity test, each latent variable has an AVE value greater than 0.50. Figure 6 depicts the final model.

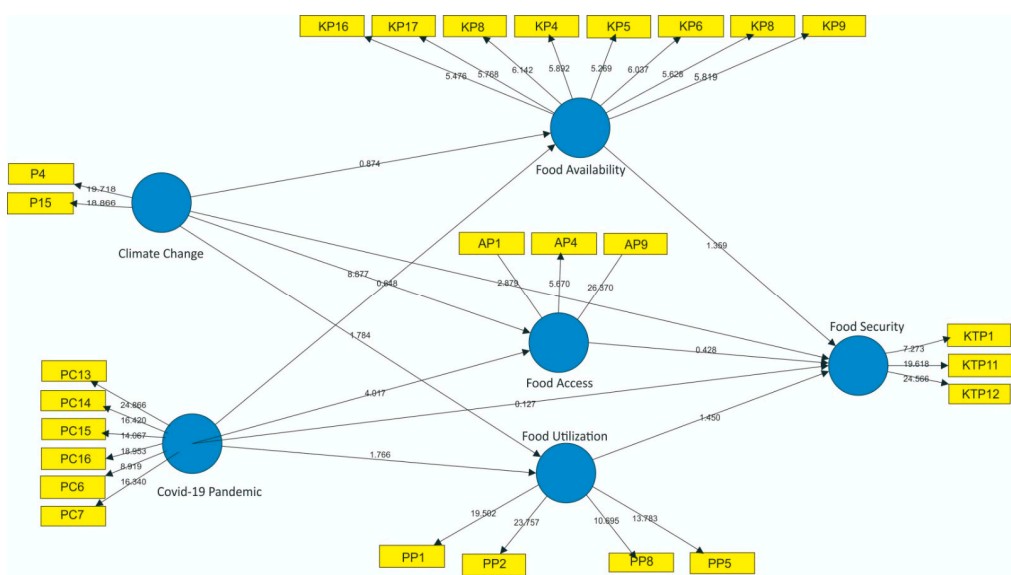

**Figure 6.** Final research model.

All latent variables in the composite reliability test are known to have a Cronbach's alpha value of 0.60, and these variables meet the requirements for the composite reliability test. This follows the concept that the research model can be accepted as valid and credible by eliminating eleven variables. The inner model is assessed by analyzing the R-square value and the path coefficient. In addition, this model uses the R-square value to determine how much influence the exogenous variables have on the endogenous variables.

The results of the convergent validity test are determined based on the principle that the measurement of a construct must have a high correlation (Joseph F. Hair et al. 2019). The convergent validity of a construct with a reflective indicator was evaluated by average variance extracted (AVE). The minimum AVE value is equal to or greater than 0.5, while an AVE value of 0.5 or more means that the construct can explain more than 50% of the item variance.

Analysis of *discriminant validity*, which is carried out to ensure that each concept of each latent model is different from other variables, refers to value cross-loading, or the *Fornell-Larcker criterion*, from the manifest variable to its latent variable. *Discriminant validity* aims to test to what extent the latent construct differs from other constructs. A high value indicates that a construct is unique and able to explain the phenomenon being measured. All latent construct values must be greater than the correlation with other constructs so that

the discriminant validity requirements in this model have been fulfilled. The calculation results of the *Fornell-Larcker criterion* for *discriminant validity* are shown in Table 5.

**Table 5.** Fornell-Larcker criterion for discriminant validity.

| Items | AP | KTP | KP | PC | PP | PI |
|---|---|---|---|---|---|---|
| AP | 0.798 | | | | | |
| KTP | 0.429 | 0.857 | | | | |
| KP | 0.259 | 0.263 | 0.805 | | | |
| PC | 0.551 | 0.317 | 0.149 | 0.816 | | |
| PP | 0.597 | 0.457 | 0.254 | 0.455 | 0.848 | |
| PI | 0.526 | 0.522 | 0.104 | 0.403 | 0.489 | 0.879 |

Source: Compiled by the author. Remarks: AP: access to food; PC: COVID-19 outbreak; KTP: food security; PI: climate change; KP: food availability; PP: food utilization.

All latent variable values in the model have findings that are greater than the correlation values of the other latent variable constructs, according to measurements using the Fornell-Larcker criterion, as shown in Table 5. As a result, this model meets the criteria for discriminant validity (J. F. Hair et al. 2019). Furthermore, it may be argued that this model is both usable and legitimate. To test and evaluate the inner model, R-square and path coefficient significance are used. The impact of exogenous variables on endogenous variables is measured using the R-square. Table 6 displays the R-square calculation's outcomes.

**Table 6.** Results of the R-Square.

| Variable | R-Square |
|---|---|
| Access to Food | 0.413 |
| Food security | 0.355 |
| Food availability | 0.025 |

Source: Compiled by the author.

According to Table 6, which is in the moderate category, the food access variable can explain the food security variable by 41.3%, and the food utilization variable can explain the food security variable by 31.9%. Food availability and food security have a 2.5% association, which is considered to be low. A 35.5% link exists between food security and the accessibility, utilization, and availability of food. There are additional elements that influence food security in this situation, including the stability of the food system.

The route coefficient significance test, which uses the bootstrap method, produced the original sample values, *p*-values, and t-statistic values to assess the research model and research hypotheses. The variables' relationships to one another are made evident by the initial sample values. A variable transforms from a negative to a positive state when it has a positive influence, and vice versa. A hypothesis test's significance can be determined by calculating the t-statistic value. The *p*-value is less than 0.05, which indicates that the hypothesis is supported. Table 7 displays the value of the route coefficient, which depicts the link between all variables.

Food security is positively impacted by access to food, with a *p*-value greater than 0.05. This shows that despite a variety of circumstances and conditions that might make it difficult to access food, residents of Cibunian Village and Purwabakti Village, Pamijahan District, will continue to make an effort to meet their food needs, so H1 is not accepted.

With a *p*-value greater than 0.05, food availability has a beneficial but small impact on food security. As a result, H2 is not accepted. This shows that food is still available for the citizens of Cibunian Village and Purwabakti Village in the Pamijahan District, though the quantity and variety have declined at the home level and in food supply facilities.

**Table 7.** Path coefficients.

| Path | Original Sample | T-Statistic | *p*-Value | Hypotheses |
|---|---|---|---|---|
| AP → KTP | 0.074 | 0.428 | 0.334 | H1 not accepted |
| KP → KTP | 0.155 | 1.359 | 0.087 | H2 not accepted |
| PC → AP | 0.404 | 4.017 | 0.000 | H3 accepted |
| PC → KTP | 0.021 | 0.127 | 0.449 | H4 not accepted |
| PC → KP | 0.128 | 0.648 | 0.259 | H5 not accepted |
| PC → PP | 0.308 | 1.765 | 0.039 | H6 accepted |
| PP → KTP | 0.184 | 1.450 | 0.074 | H7 not accepted |
| PI → AP | 0.363 | 3.377 | 0.000 | H8 accepted |
| PI → KTP | 0.368 | 2.879 | 0.002 | H9 accepted |
| PI → KP | 0.052 | 0.374 | 0.354 | H10 not accepted |
| PI → PP | 0.365 | 2.784 | 0.003 | H11 accepted |

Source: Compiled by the author.

With a *p*-value less than 0.05, the COVID-19 outbreak has a positive and significant influence on the food access variable, indicating that the COVID-19 outbreak has caused a fall in the income of residents in Cibunian Village and Purwabakti Village. Furthermore, the COVID-19 outbreak has raised food prices, impeded food distribution, and forced the closure of food supply facilities. In the event of a COVID-19 outbreak, there is a policy of restricting community activities that influence the food delivery system, so H3 can be accepted.

With a *p*-value greater than 0.05, the COVID-19 outbreak has a positive but insignificant influence on food security. This demonstrates that, despite the COVID-19 outbreak, Pamijahan residents continue to consume food, but the amount consumed has dropped, particularly consumption of animal food, causing family members, including toddlers, to feel unwell more frequently. As we know, the output of food security is good nutritional and health status, which is shown in the monthly increase in the weight of children under five and the consumption of food by the needs for an ideal healthy life, resulting in H4 being accepted.

The COVID-19 outbreak has a positive but insignificant effect on food availability, with a *p*-value of more than 0.05. This shows that food is still available at the household level and in food supply facilities, but the amount and type have decreased. Apart from that, the COVID-19 outbreak has also caused the reserve fund to buy food to decrease, so H5 was not accepted.

The COVID-19 outbreak had a beneficial and substantial effect on food utilization, with a *p*-value less than 0.05. Because of the COVID-19 outbreak, the variety and amount of food consumed have decreased, while the use of rice fields for food sources has increased, allowing H6 to be accepted.

Food utilization has a positive but insignificant effect on food security, with a *p*-value greater than 0.05. This suggests that residents of Cibunian Village and Purwabakti Village continue to eat every day, but the frequency and kind of food consumed have decreased. Furthermore, the quality of drinking water has deteriorated, causing H7 to be accepted.

Climate change has a favorable and significant impact on food access, with a *p*-value less than 0.05. This demonstrates that floods caused by climate change reduce the amount of money available for food purchases. Flooding also forced food providers to change modes of conveyance due to a lack of road access. Furthermore, during flood conditions, food supply facilities such as booths and shops are closed, so H8 can be accepted.

With a *p*-value less than 0.05, climate change has a positive and substantial effect on food security. The nutritional and health status of a community reflects the region's food security situation. This suggests that floods caused by climate change have made residents of Cibunian Village and Purwabakti Village sicker more frequently; hence, H9 is accepted.

With a *p*-value greater than 0.05, climate change has a positive and substantial effect on food availability. This demonstrates that food is still available during floods caused

by climate change, but the amount and type at the household level, as well as the food provider's advice, are reduced, hence H10 is refused.

### 3.3. Public Perception of the Effects of Climate Change and the COVID-19 Outbreak on Food Security in Pamijahan District

According to Watts (2009), sustainable development will be successful if all stakeholders, including the community, work together as a single organism. It is critical to understand the perspectives of those who are touched by the events that occur, as well as local knowledge, to reduce existing problems. As a result, this study employs a sequential explanatory mixed-methods design (John W. Creswell 2014), which includes both quantitative and qualitative components. The qualitative technique based on in-depth interviews seeks to ascertain the community's opinion on the impact of climate change and the COVID-19 outbreak on food security and existing local ecological knowledge.

Based on in-depth interviews, flood-affected residents endure fear and trauma. Residents stated that the flood disaster had a greater impact on the economy and family food availability than the COVID-19 outbreak because the flood damaged rice fields and fish ponds, which are also sources of livelihood and food for residents, as well as transportation for food providers. Rising temperatures, erratic weather, and high-intensity rains causing floods and landslides are all symptoms of climate change. During the COVID-19 outbreak, no inhabitants were confirmed positive for COVID-19, although health precautions were put in place to avoid COVID-19 transmission.

Flooding of fish ponds and rice fields caused crop failure among the inhabitants, resulting in a decreased food supply. The water also destroyed food storage facilities (refrigerators, cupboards, and other items), leaving inhabitants without food. The flood devastated the road transportation infrastructure, isolating the town and making it difficult for locals to obtain rice and other basic components. The state of food supplies was likewise unstable for a few months following the disaster. Those who could previously buy 30 kg of rice can now only buy 10 kg of rice, while those who could previously buy 1 kg of eggs can now only buy 2 eggs.

Flood-affected residents saw a greater than 50% loss in income, affecting their capacity to obtain food. Flooding cost 68.8% of respondents their jobs as farm laborers. Furthermore, the flood destroyed highways, resulting in a scarcity of some foodstuffs.

A decreased income forces residents to cut their food budget, reducing the frequency of meals for flood-affected residents, which is not in accordance with a balanced nutritional diet. Aside from that, locals reported that following the floods and the COVID-19 outbreak, residents, particularly toddlers, became more easily ill. This has the potential to generate nutritional and public health issues if not handled properly.

According to in-depth interviews, floods caused by climate change and the COVID-19 outbreak have had an impact on all facets of food security in food-surplus but disaster-prone countries. Farm laborers provide the majority of the livelihoods of those affected by the crisis, according to (The Economist Group 2022), and farmers are the most susceptible group because of climate change and the COVID-19 outbreak. Agriculture is fundamentally reliant on the environment. While agriculture is crucial for producing food and providing nutrients for human health, it can also have an impact on the environment by polluting air and water and generating greenhouse emissions (Gilbert 2012).

Agriculture, food, public health, and climate change are all interconnected in different ways (Lam et al. 2017; Ramachandran et al. 2020). Declining environmental quality can have an impact on public health and raise health care spending, which in turn has an impact on agricultural and food production (Wu et al. 2016). Environmentally friendly agricultural and food systems, on the other hand, can reduce GHG emissions, improve public health, and raise the capacity and output potential of future agricultural systems (Barbier 2020).

For the establishment of an environmentally friendly agricultural and food system, local ecological knowledge and resources must be studied (Lam et al. 2017). Based on

conversations with agricultural extension workers, traditional leaders, and local farmers in Pamijahan District, they continue to adopt sustainable farming practices based on local ecological knowledge. Agricultural activity is not simply a means of meeting human needs but also of drawing closer to the Creator. Before beginning the planting process, Pamijahan local community/farmers hold a scattering meeting attended by the Association of Farmers Groups (Gapoktan), neighborhood associations, village heads, and village elders to determine the planting time and location based on constellations.

Local farmers control pests and diseases by rotating planting kinds, trading seeds among farmers, and using natural pesticides. Local farmers believe that in growing rice, everything in the food chain, including pests and animals, has a role, so they do not use chemicals to manage them. This is consistent with the concepts of sustainable agriculture, which include adaptive agriculture that continues to grow, remains functional, is resistant to stress, becomes productive, uses resources efficiently, and balances sustainability goals at all scales (Mucharam et al. 2022).

Citizens in Pamijahan District have a history of preserving the harvest (rice) in a specific room called "Goah" or Leuit to assure the availability of food stocks. Rice storage in Goah/Leuit is being encouraged anew as the Pamijahan community abandons the habit. Goah/Leuit protects food security, especially in times of calamity or crop failure (Kusdiwanggo 2020).

Respondents in this study were people in Pamijahan District, Bogor Regency, who were devastated by floods and landslides on 22 June 2022. There were 64 households that responded. Respondent factors such as age, occupation, education, and income were examined. Table 8 shows further features of the respondents.

**Table 8.** Respondent criteria.

| Characteristics | Criteria | Percentage (%) |
|---|---|---|
| Age (year) | 31–40 | 43.8 |
| | <50 | 21.9 |
| | 21–30 | 20.3 |
| | 41–50 | 14.1 |
| Job | Agricultural labor | 70.2 |
| | Self-employed | 10.9 |
| | Farmer | 9.4 |
| | Trader | 3.1 |
| | Driver | 1.6 |
| | Debt collector | 1.6 |
| | Motorcycle driver | 1.6 |
| | Teacher | 1.6 |
| Level of education | Elementary School | 76.6 |
| | Junior High School | 12.5 |
| | Senior High School | 7.8 |
| | Bachelor Degree | 3.1 |
| Income per month (IDR) | Rp. 1,000,000–1,500,000 | 32.8 |
| | Rp. 500,000–1,000,000 | 20.3 |
| | Rp. 2,000,000–3,000,000 | 18.8 |
| | Rp. 1,500,000–2,000,000 | 17.2 |
| | Rp. 0–500,000 | 9.4 |
| | Rp. >3,000,000 | 1.6 |

Source: Compiled by the author.

The characteristics of the respondents in this study were dominated by the age range of 31–40 years, namely 43.8%. This age range covers the productive age, which has the potential to increase resilience to face disaster threats due to climate change and non-natural disaster threats. Farm laborers are the dominant occupation of the respondents in this study, namely 68.8%. Farm laborers are people who receive wages by working in



other people's gardens or fields. The income of 32.8% of the respondents in this study amounted to Rp. 1,000,000–1,500,000 per month, which is still far below the Bogor Regency regional minimum wage of 4,217,206. The educational level of 76.6% of the respondents is elementary school.

### 3.4. Managerial Implications

The study's managerial implications are being implemented to establish food security in disaster-prone communities in Pamijahan District, Bogor Regency. Based on the findings of the food security situational analysis and qualitative descriptive analysis, the aspects most affected by climate change, specifically floods and the COVID-19 epidemic, are food availability and food access. The pathway analysis results also show that in order to mitigate the impacts of climate change and the COVID-19 epidemic, it is important to improve food availability and access.

To achieve food availability that fulfills the needs of each individual, the people of Pamijahan District may reestablish the tradition of storing rice/food crops in "Goah/Leuit" to guarantee food availability in natural and non-natural disaster situations. Furthermore, the community must re-implement sustainable agriculture by combining local ecological knowledge (being environmentally friendly, not greedy, and using agriculture to get closer to the Creator) with the most current advances in science and technology (using climate and weather information and disaster information in agricultural planning).

Stable food availability can be achieved by bringing food closer to the community, where it is not affected by price changes, food delivery problems caused by disasters, or decreasing incomes. A potential strategy is to use the grounds around the house for food (Boyacı-Gündüz et al. 2021).

Adaptation to disaster mitigation due to climate change is accomplished by the use of an agroforestry pattern in the implementation of sustainable agriculture (Alfatikha et al. 2020). This involves combining agricultural and forestry crops to increase resilience and reduce the danger of landslides, floods, and droughts.

## 4. Discussion

Climate change through increasing temperatures has an impact on rising sea levels, high rainfall intensity, and drought, which threaten food security and nutrition through interrelated impacts on land for agriculture, the growth of crops, the survival of sources of animal food, and labor productivity in agriculture (Hendriks et al. 2022). The average temperature in Pamijahan District in the period 1991–2022 increased by 0.60 °C. This temperature increase is still within normal limits, referring to the Paris Agreement (UNFCC 2015), which states that the maximum limit for the global average temperature rise internationally is 1.5 °C. Increasing temperatures cause changes in distribution and rainfall and increase the potential for extreme climates.

At least one extreme climate event occurred between 1991 and 2022. A longer dry season resulted in drought, which was positively related to disaster events in Pamijahan District in particular. Temperature changes also affect seasonal shifts, such as a shorter rainy season with higher rainfall (floods) and a longer dry season. Bogor Regency is included in the most disaster-prone areas in West Java; according to provincial disaster-prone index data for 2011, it ranked fifth overall.

Torrential rains caused severe flooding and landslides in Pamijahan District, Bogor Regency, on 22 June 2022. These events caused three people to die, displaced more than 335 people, and destroyed more than 281 houses, schools, health facilities, and infrastructure (Regional Disaster Management Agency 2022). Floods in Pamijahan District caused disturbances throughout the food system, including damage to food supply infrastructure, namely rice fields, fields, fish ponds, food storage, transportation, and markets. In addition, access to food and clean water is also hampered (Hendriks et al. 2022). Because the majority of the affected residents were agricultural laborers, the floods that damaged rice fields and fish ponds also killed the residents' sources of income and livelihoods.

Floods, landslides, and strong winds are the three main hydrometeorological disasters in Pamijahan District. The probability of food insecurity in an area will increase if disasters occur frequently there (Baliwati 2019). Food insecurity can also be understood as a situation in which access to and consumption of food in a place, community, or household are insufficient to meet everyone's physiological needs for survival and growth. Food insecurity can occur simultaneously with certain events.

Using two analytical approaches, information on food insecurity and its causes is examined. Food insecurity can be classified as chronic or transient, depending on the cause (Sowe et al. 2015). This study investigated transient and chronic food insecurity using an analysis of the Food and Nutrition Awareness System (SKPG) and the Food Security and Vulnerability Map (FSVA).

Based on the findings of the FSVA analysis, the condition of food security in the Cibunian and Purwabakti villages in 2022 will drop to Priority 2 (somewhat vulnerable to food insecurity). From the end of 2019 until the end of 2022, the COVID-19 outbreak occurred, which attacked aspects of food access and significantly decreased the income and welfare of the population, resulting in a decrease in food purchasing power (Béné et al. 2021). The disruption caused by the COVID-19 outbreak has affected the poor and other marginalized groups, especially those with low purchasing power (Roubík et al. 2023).

As a result of crop failure, there is an increase in food insecurity in terms of food availability. Natural disasters that damage agricultural land and intensify the existence of creatures that disturb plants are one of the factors causing these conditions (Nuraisah and Kusumo 2019).

In addition to the unresolved COVID-19 disaster, in 2022 there will be flash floods, which will affect the food supply in Cibunian Village and Purwabakti Village. It was emphasized by (Rasul 2021) that the COVID-19 outbreak has largely had an impact on the components of food access related to decreased purchasing power and food shortages as a result of widespread social restrictions. In addition, in 2022, the villages of Cibunian and Purwabakti will experience flash floods, which will hinder access to logistics and hamper food distribution.

The percentage of under-fives in Pamijahan District is a benchmark for food utilization. Malnutrition in toddlers during the COVID-19 outbreak can be caused by several factors, including a decrease in services due to the restrictions resulting from the outbreak (Yuwansyah et al. 2021). In the 2018–2022 period, the frequency of underweight toddlers increased by 8.3% in Pamijahan District.

Five of the eleven feasible hypotheses were confirmed by this study:

1. The variable food availability has been positively and significantly impacted by the COVID-19 outbreak; this shows that the outbreak has reduced household income in Cibunian Village and Purwabakti Village. The COVID-19 outbreak in Cibunian Villagae and Purwabakti Village has also had an impact on rising food prices, difficulties in food delivery, and the closure of food supply facilities (shops/basketball stalls). A policy to impose local operational restrictions also had an impact on the food distribution system during the COVID-19 outbreak situation.

2. Utilization of food has been positively and significantly impacted by the COVID-19 outbreak. This shows that the COVID-19 outbreak has resulted in a decrease in overall food consumption and an increase in the use of carrots as a food source.

3. Access to food is positively and significantly affected by climate change. This shows how catastrophic floods caused by climate change reduce the amount of money available to buy food and change the mode of transportation for those who deliver food since the floods cut off access to roads. Additionally, during flood situations, food supply facilities such as booths and shops are closed.

4. Food security is positively and significantly affected by climate change. The nutritional and health status of a community reflects the level of food security in that location. This shows that flooding due to climate change has made people sick more often in the Cibunian and Puwabakti villages.

5.  Food use is positively and significantly affected by climate change. It shows how disasters caused by climate change reduce the quality of food and water.

## 5. Conclusions and Implications

### 5.1. Conclusions

The condition of food security in Pamijahan District, especially Cibunian Village and Purwabakti Village, meets the criteria of food insecurity for the period 2017–2022, according to FSVA analysis from the aspects of food availability, access, and utilization. Additionally, according to SKPG analysis, there is an increase in food insecurity of 8.3% for food availability, an increase of 33.3% for food access, and an increase of 8.3% for food utilization.

This study found a positive and substantial relationship between the COVID-19 outbreak variables, food access variables, and food use variables. In addition, there is a positive and robust relationship between climate change variables and food access, food use, and food security.

### 5.2. Limitations and Future Research

The constraints of this study are those imposed by the author, notably for the responding subjects, who are residents of two villages in Pamijahan District who have been directly affected by natural catastrophes and the COVID-19 outbreak. Second, this study is based on the perceptions of individual respondents who were intimately involved in the natural tragedy and the COVID-19 outbreak when food insecurity occurred concurrently.

Third, this study analyzes the direct influence of the correlation of all dependent variables in the developed food security model. Subsequent research can replicate all of the model variables that have been developed by involving the entire larger area over a longer period of time and by adding the dependent variable of the presence of digitization of food products that is getting closer to producers and consumers by comparing it with different regions within one province.

### 5.3. Policy Implications

It is critical to incorporate long-term sustainability into short-term policy decisions while developing policies and strategies. However, to ensure that short-term measures create long-term advantages, strategic thinking and a rigorous assessment of long-term policy alternatives and investment plans are required. Policies and plans should be connected with local ecological demands and knowledge, both short- and long-term.

Climate change and the COVID-19 outbreak present numerous challenges to global and regional food security. This study explores local ecological knowledge for establishing food security in disaster zones as well as the impact of climate change and the COVID-19 outbreak on food security. As a result, adaptation to climate change mitigation and the COVID-19 outbreak must be tackled by all stakeholders, especially local organizations or indigenous peoples who combine local ecological knowledge with the most recent scientific and technological breakthroughs.

### 5.4. Policy Recommendations

Policy recommendations:

1.  Improving the process of developing social capital and other resources in order to build a disaster-resilient and food-independent society based on local ecological knowledge combined with scientific findings and cutting-edge technology.
2.  Investigating and protecting local ecological knowledge through the use of the law and the establishment of non-commercial cultural places.
3.  Improving farmers' access to financial independence by developing an environmentally and health-friendly circular economy.
4.  The government should add sustainable living information to formal and non-formal education.

5. The government should invest in agriculture research and innovation that is both sustainable and health-friendly.

**Author Contributions:** Conceptualization, F.A.; methodology, J.I., F.A. and E.R.; investigation, F.A.; resources, F.A.; writing—original draft preparation, F.A.; writing—review, F.A., J.I. and E.R.; writing—editing, E.R. and J.I. All authors have read and agreed to the published version of the manuscript.

**Funding:** This research received no external funding.

**Institutional Review Board Statement:** Not applicable.

**Informed Consent Statement:** Not applicable.

**Data Availability Statement:** Not applicable.

**Acknowledgments:** Directorate of Research and Community Service (DRPM), Universitas Padjadaran.

**Conflicts of Interest:** The authors declare no conflict of interest.

## Appendix A

**Table A1.** Survey (the original survey used with the participants was translated into this version).

| Latent Variable | Manifest Variable | Code | Value | | | | |
|---|---|---|---|---|---|---|---|
| | | | 1 | 2 | 3 | 4 | 5 |
| Climate Change | The current temperature/environment is hotter than it was 10–20 years ago | PI1 | | | | | |
| | The rainy season lasts longer now than 10–20 years ago | PI2 | | | | | |
| | The dry season lasts longer than 10–20 years ago | PI3 | | | | | |
| | Rainy season patterns are easier to predict today than they were 10–20 years ago | PI4 | | | | | |
| | The dry season pattern is easier to predict now than it was 10–20 years ago | PI5 | | | | | |
| | Rainfall more often now than 10–20 years ago | PI6 | | | | | |
| | There has been a lot of deforestation and land conversion over the past 10–20 years | PI7 | | | | | |
| | Landslides are happening more frequently now than 10–20 years ago | PI8 | | | | | |
| | Tornado disasters are happening more frequently now than 10–20 years ago | PI9 | | | | | |
| | Flood disasters are more common now than 10–20 years ago | PI10 | | | | | |
| | Crop failure due to disasters is more common now than 10–20 years ago | PI11 | | | | | |
| | Plant pests were more common 10–20 years ago | PI12 | | | | | |
| | Droughts are more common today than they were 10–20 years ago | PI13 | | | | | |

<p style="text-align:center">**Table A1.** *Cont.*</p>

| Latent Variable | Manifest Variable | Code | Value | | | | |
|---|---|---|---|---|---|---|---|
| | | | 1 | 2 | 3 | 4 | 5 |
| COVID-19 | Some villagers who have been confirmed infected by COVID-19 | PC1 | | | | | |
| | There are restrictions on activities during the COVID-19 | PC2 | | | | | |
| | There is independent isolation for residents positive for COVID-19 | PC3 | | | | | |
| | The head of the family often gets sick | PC4 | | | | | |
| | Family members often experience pain | PC5 | | | | | |
| | There are anticipatory steps to prevent disease | PC6 | | | | | |
| | There are efforts by the village government to anticipate the threat of disease outbreaks | PC7 | | | | | |
| | There is cooperation in handling sick residents | PC8 | | | | | |
| | There is a movement for the consumption of healthy, nutritious, and balanced food | PC9 | | | | | |
| | The family always does regular exercise | PC10 | | | | | |
| | Online school | PC11 | | | | | |
| | There is a health protocol in every social activity in the village | PC12 | | | | | |
| | Villagers wear masks when going out | PC13 | | | | | |
| | Villagers always wash their hands | PC14 | | | | | |
| | Villagers keep their distance | PC15 | | | | | |
| | Villagers carry out vaccines | PC16 | | | | | |
| Food Availability | Climate Change | | | | | | |
| | Climate change causes floods | KP1 | | | | | |
| | Food production decreased due to the flood disaster | KP2 | | | | | |
| | The amount of food availability in the market is reduced after the flood disaster due to climate change | KP3 | | | | | |
| | The type of food availability in the market is reduced after the flood disaster due to climate change | KP4 | | | | | |
| | The amount of food availability in households has increased after the floods caused by climate change | KP5 | | | | | |
| | The type of household food availability has increased after the flood disaster due to climate change | KP6 | | | | | |
| | Household food stores decreased after the flood disaster | KP7 | | | | | |
| | There is a reserve fund to buy food after catastrophic floods due to climate change | KP8 | | | | | |
| | There is a village food barn after the flood disaster caused by climate change | KP9 | | | | | |
| | COVID-19 | | | | | | |
| | Food production has decreased due to COVID-19 | KP10 | | | | | |
| | The amount of food availability in the market has decreased after the COVID-19 | KP11 | | | | | |
| | The type of food availability in the market has decreased after the COVID-19 | KP12 | | | | | |

**Table A1.** *Cont*.

| Latent Variable | Manifest Variable | Code | Value | | | | |
|---|---|---|---|---|---|---|---|
| | | | 1 | 2 | 3 | 4 | 5 |
| | The amount of food availability in households has increased after the COVID-19 | KP13 | | | | | |
| | The type of household food availability has increased after the COVID-19 | KP14 | | | | | |
| | Household food stocks are reduced after the COVID-19 | KP15 | | | | | |
| | There is a reserve fund to buy food after the COVID-19 | KP16 | | | | | |
| | There is a village food barn after the COVID-19 | KP17 | | | | | |
| | Climate Change | | | | | | |
| | Family income decreased after the flood disaster | AP1 | | | | | |
| | Income decreased after the flood disaster | AP2 | | | | | |
| | The flood disaster caused food distribution to be hampered | AP3 | | | | | |
| | Changes in means of transportation of foodstuffs after the flood disaster | AP4 | | | | | |
| | The flood disaster caused a decrease in the cost of storing food | AP5 | | | | | |
| | Food prices increased after the floods | AP6 | | | | | |
| Food Access | Increased food prices cause food shortages in households | AP7 | | | | | |
| | Many roads were damaged by the floods | AP8 | | | | | |
| | Stores/stalls providing food are closed due to the flood | AP9 | | | | | |
| | COVID-19 | | | | | | |
| | Family income has decreased after COVID-19 | AP10 | | | | | |
| | Income reduced after COVID-19 | AP11 | | | | | |
| | COVID-19 has hampered food distribution | AP12 | | | | | |
| | Changes in food transportation equipment after COVID-19 | AP13 | | | | | |
| | COVID-19 causes a decrease in the cost of storing food | AP14 | | | | | |
| | Food prices increased after the COVID-19 | AP15 | | | | | |
| Food Access | Increased food prices cause food shortages in households | AP16 | | | | | |
| | Stores/stalls providing food are closed due to the COVID-19 | AP17 | | | | | |
| | Climate Change | | | | | | |
| | There was a decrease in the quality of food after the flood disaster | PP1 | | | | | |
| | Flood disaster causes foodstuffs/food to be cleaner and safer | PP2 | | | | | |
| | Flood disaster caused foodstuffs/food to be stored longer | PP3 | | | | | |
| | The amount of food eaten became more after the flood disaster | PP4 | | | | | |
| | The frequency of eating is more frequent after the flood disaster | PP5 | | | | | |
| Food Utilization | Utilization of coral reefs for food crops increased after the flood | PP6 | | | | | |
| | Leftover food decreased after the flood disaster | PP7 | | | | | |
| | The frequency of cooking decreased after the flood disaster | PP8 | | | | | |
| | COVID-19 | | | | | | |
| | There has been a decline in the quality of drinking water after COVID-19 | PP9 | | | | | |
| | There has been a decline in food quality after COVID-19 | PP10 | | | | | |

**Table A1.** *Cont.*

| Latent Variable | Manifest Variable | Code | Value | | | | |
|---|---|---|---|---|---|---|---|
| | | | 1 | 2 | 3 | 4 | 5 |
| | COVID-19 causes foodstuffs/food to be cleaner and safer | PP11 | | | | | |
| | COVID-19 causes foodstuffs/food to be stored longer | PP12 | | | | | |
| | The type of food has become more varied COVID-19 | PP13 | | | | | |
| | The amount of food eaten becomes more | PP14 | | | | | |
| | The amount of food consumption is higher after COVID-19 | PP15 | | | | | |
| | The use of coral reefs for food crops increased after COVID-19 | PP16 | | | | | |
| | The application of hydroponic cultivation has increased after the outbreak | PP17 | | | | | |
| | Food leftovers have decreased after the COVID-19 Outbreak | PP18 | | | | | |
| | Cooking frequency decreased after the COVID-19 Outbreak | PP19 | | | | | |
| | COVID-19 causes patterns of food consumption | PP20 | | | | | |
| | Climate Change | | | | | | |
| | Flood disasters cause you to get sick more often | KTP1 | | | | | |
| | Flood disasters due to climate change cause your family members to get sick more often | KTP2 | | | | | |
| | I feel all of my suits getting bigger after flood disaster | KTP3 | | | | | |
| | The weight of your family members decreased after the flood disaster | KTP4 | | | | | |
| | Food availability at household decreased after the flood disaster | KTP5 | | | | | |
| Food Security | Food consumption increased after the flood disaster | KTP6 | | | | | |
| | Consumption of animal and vegetable side dishes increased after the flood | KTP7 | | | | | |
| | Vegetable consumption increased after the flood | KTP8 | | | | | |
| | Your toddler's body weight went down after the flood | KTP9 | | | | | |
| | Your toddler is often sick after the flood | KTP10 | | | | | |
| | COVID-19 | | | | | | |
| | COVID-19 cause you to get sick more often | KTP11 | | | | | |
| | COVID-19 cause your family members to get sick more often | KTP12 | | | | | |
| | I feel all of my suits getting bigger after COVID-19 | KTP13 | | | | | |
| | The weight of your family members decreased after the COVID-19 | KTP14 | | | | | |
| | Food availability at household decreased after COVID-19 | KTP15 | | | | | |
| Food Security | Food consumption increased after the COVID-19 | KTP16 | | | | | |
| | Consumption of animal and vegetable side dishes increased after COVID-19 | KTP17 | | | | | |
| | Vegetable consumption increased after COVID-19 | KTP18 | | | | | |
| | Your toddler's body weight went down after COVID-19 | KTP19 | | | | | |
| | Your toddler is often sick after the COVID-19 | KTP20 | | | | | |

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
