# Peer review of "The Impact of COVID-19 and Climate Change on Food Security in Pamijahan District, Bogor Regency"

_economies, doi:10.3390/economies11110271_

Round 1
Reviewer 1 Report
Journal
Economies (ISSN 2227-7099)
Manuscript ID
economies-2530145
Type
Article
Title
Impact of The Covid-19 Pandemic and Climate Change on Food Security in Pamijahan District, Bogor Regency
The article presents an investigation into food security in the Pamijahan District, with a particular focus on the influence of climate change and the Covid-19 epidemic during the period 2017-2022. The authors aim to present how the availability, accessibility, and consumption of food are affected by these factors using a mixed-method qualitative and quantitative approach.
Overall, the study attempts to address a relevant and pressing issue, but there are several critical points that need to be addressed for the paper to be considered scientifically rigorous and informative:
Lack of Clarity in the Introduction as the introduction lacks clarity and conciseness, making it difficult for readers to fully understand the research objective and the context of the study. The authors should provide a clear and comprehensive overview of the current state of food security in disaster-prone areas and the significance of investigating the impact of climate change and the Covid-19 pandemic on food security.
Methodological Weaknesses, as the paper mentions the use of mixed-method qualitative and quantitative analysis, but the specific details about the methods used are not adequately described in the abstract. Then in the methods section, some typos are reducing the trustworthiness. It is crucial to provide a clear and comprehensive outline of the research design, sampling techniques, data collection, and analytical procedures. Without this information, it is challenging for readers to evaluate the robustness and validity of the study. Similarly like statements “processed from the relevant agencies” do not provide adequate potential for replicability.
The study's findings are presented without sufficient context and interpretation – meaning that the discussion is not strong enough. The authors should provide a more detailed discussion of the results, including a clear explanation of how the COVID-19 pandemic and climate change impact food security in the region. Additionally, it is essential to present the findings in a more organized and structured manner to facilitate readers' understanding.
The also article mentions "varied models" of food security, but there is maybe not enough comparison and discussion of these models in the context of the study's findings. A critical analysis of how existing food security models relate to the specific conditions in target area/district could enhance the paper's scientific significance.
Some studies to potentially discuss with:
https://www.sciencedirect.com/science/article/pii/S2667010021000068
https://www.frontiersin.org/articles/10.3389/fvets.2020.578508/full
https://www.sciencedirect.com/science/article/pii/S2352550922000082
https://www.bmj.com/content/378/bmj-2022-071534
https://www.sciencedirect.com/science/article/pii/S0308814621018367
In conclusion, while the study's objective is relevant and addresses an important issue, there are significant weaknesses that need to be addressed to enhance the scientific rigor and clarity of the article. By providing more comprehensive details on the methodology, improving data presentation and interpretation, and addressing the limitations of the study, the authors can significantly improve the quality and impact of their research.
In addition, the policy implication are not really policy implications. There are some recommendations. However, implications and recommendations are two different issues.
Also – it is either Coronavirus disease 2019 or COVID-19.
Moderate editing of English language required
Author Response
Dear Prof
Thank you very much for your attention, I've attached files with to response your comments.
I sincerely appreciate it. You have made very wise suggestions. I believe that your suggestions have improved our article.
Sincerely,
Author

Reviewer 2 Report
Based on the article received, I feel that the manuscript could be reconsidered for publication after considering all the major revisions attached below:
1. In the introduction section, please clarify the justification for this research. The introduction section is too short and fails to show the research gap and novelty of this research. The problem statement is not well organized and is not clear to the reader. There is no methodological explanation or reason for choosing the partial least square structural equation simulation (PLS-SEM). Also, there is a considerable gap between the novelty of the research and the discussion of existing studies. Please make it as elaborate as you can. Substantial changes and revisions are required in the introduction section. Please make a story and try to find out the importance of this research.
2. Please add a study area map in the description of the research area section.
3. Please develop a conceptual framework for better understanding your study.
4. Please add the theoretical background of using a partial least square structural equation simulation (PLS-SEM).
5. Shorten this part and move it to the justification of the background of your study part. 3.1. General condition of Pamijahan District; 3.1.1 Climate Conditions in Pamijahan District 1990-2022 and 3.1.2 Hydrometeorological Disaster Development in Pamijahan District. This is not your findings. Also requested to make it concise and specific.
6. You are requested to add a descriptive table.
7. Please add a correlation matrix table.
8. Please add a variable description table.
9. Please revise the reference style and format.
Finally, I would like to say that the research topic is important in the present context. Research ideas are also good, but you should focus on the consistency of your writing and the selection of models and variables.
Extensive English editing is required.
Author Response

(The authors gave the same response as above.)

Round 2
Reviewer 1 Report
Dear authors,
I can see that you significantly improved your paper.
Several minor comments:
Provide all equations in "Equation function".
I recommend to discuss two additional recent papers from 2023 to show, that you reviewed also the most recent literature on the topic:
https://link.springer.com/article/10.1007/s11356-023-25714-1
https://link.springer.com/article/10.1186/s12992-023-00952-7
Figure 3 and 7 needs to have better quality.
Policy recommendations provide in form of bullet points.
With regards,
The Reviewer
Moderate editing of English language required
Author Response
Dear Prof
Thank you very much for your attention to us.
We believe your comments and suggestion are great to make our article becomes excellent. You may see our cover letter report in the attachment

Reviewer 2 Report
1. Please revise the introduction section. The justification part is still vague and does not properly address your research theme.
2. The conceptual framework is not well connected. Please revise accordingly. It should be well connected.
3. The theoretical background of the model is not right. Please read some good-quality journals to understand the theoretical background of a model.
4. The results section needs to be improved. You should discuss previous research findings.
5. The conclusion and policy implications section should be very specific. It should be based on your key findings and findings-based policy recommendations.
6. You should shorten your write-up rather than elaborate on it all.
7. Extensive grammatical correction is required.
8. Study area map should be developed using ArcGIS software rather than just copied and pasted.
Extensive grammatical correction is required.
Author Response

(The authors gave the same response as above.)

Round 3
Reviewer 2 Report
Thank you so much for considering all of my comments. Your paper seems better than the submitted version. I appreciate your efforts. Best of luck.